# Joint Data-Task Generation for Auxiliary Learning

**Hong Chen[1], Xin Wang[1,2]\*, Yuwei Zhou[1], Yijian Qin[1], Chaoyu Guan[1], Wenwu Zhu[1,2]\***

[1]Department of Computer Science and Technology, Tsinghua University
[2]Beijing National Research Center for Information Science and Technology, Tsinghua
{h-chen20,zhou-yw21,qinyj19,guancy19}@mails.tsinghua.edu.cn
{xin_wang,wwzhu}@tsinghua.edu.cn

## Abstract

Current auxiliary learning methods mainly adopt the methodology of reweighing losses for the manually collected auxiliary data and tasks. However, these methods heavily rely on domain knowledge during data collection, which may be hardly available in reality. Therefore, current methods will become less effective and even do harm to the primary task when unhelpful auxiliary data and tasks are employed. To tackle the problem, we propose a joint data-task generation framework for auxiliary learning (DTG-AuxL), which can bring benefits to the primary task by generating the new auxiliary data and task in a joint manner. The proposed DTG-AuxL framework contains a joint generator and a bi-level optimization strategy. Specifically, the joint generator contains a feature generator and a label generator, which are designed to be applicable and expressive for various auxiliary learning scenarios. The bi-level optimization strategy optimizes the joint generator and the task learning model, where the joint generator is effectively optimized in the upper level via the implicit gradient from the primary loss and the explicit gradient of our proposed instance regularization, while the task learning model is optimized in the lower level by the generated data and task. Extensive experiments show that our proposed DTG-AuxL framework consistently outperforms existing methods in various auxiliary learning scenarios, particularly when the manually collected auxiliary data and tasks are unhelpful.

## 1 Introduction

Auxiliary learning aims to improve the model generalization ability on the primary task with the help of related auxiliary tasks [1, 2]. This learning paradigm has been widely adopted and has shown its effectiveness in various areas, like image classification [3], recommendation [4] and reinforcement learning [5, 6]. Different auxiliary tasks are often chosen manually according to the primary task, e.g., [7] utilize the task of visual attribute classification to help the fine-grained bird classification and [4] improve the click conversion rate prediction with the help of click-through rate prediction task.

Most existing works utilize the auxiliary information by first reweighing the losses of the auxiliary data and tasks, and then use the sum of the weighted losses together with the primary loss to optimize the task learning model. The weights are employed to balance the primary loss and the auxiliary losses to avoid negative auxiliary transfer, which are tuned with HPO tools [4, 3, 8]. More recent works [9, 6, 10, 1, 11] propose to dynamically weigh different auxiliary losses during training.

However, the existing methods require that there exists beneficial information in the auxiliary data and tasks, so that the beneficial loss terms can be selected through the loss reweighing process. This condition cannot always be satisfied because whether the auxiliary data or task is beneficial depends on many factors including *the chosen auxiliary task*, *the scale of primary task dataset* and *the selected*

---

\*Corresponding Authors.

*learning model for the tasks* [1, 11], making it difficult to manually collect the beneficial auxiliary data and task via prior knowledge. Therefore, existing approaches may adopt useless auxiliary information and finally do harm to the primary task when the involved auxiliary data and tasks are improperly collected, as observed in [1, 12]. Although [2] propose to generate fine-grained auxiliary classification tasks, this method can be only applied to the classification primary task. Additionally, they only generate the new task without considering new data, while the data-level information is pointed out to be an important factor in auxiliary learning [11].

To address the problem, in this paper, we propose to simultaneously generate the auxiliary data and task for auxiliary learning in a joint manner. However, there are three challenges for the joint generation. First, it is challenging to design a generic framework that can accommodate various tasks with different inputs. This is because the types of data and tasks are quite diversified in different auxiliary learning scenarios, e.g., the primary task of image classification takes visual images as input and outputs categorical labels for classification [7], while the primary task of rating prediction for recommendation takes tabular data as input and outputs numeric labels for regression [4]. Second, it is challenging to develop a generation framework that is expressive enough to produce beneficial data and tasks for the primary task. Finally, to guarantee the jointly generated auxiliary data and task are beneficial to primary task, how to effectively optimize the parameters in the generation framework is a challenging problem as well.

To tackle these challenges, we propose a joint Data-Task Generation framework for Auxiliary Learning (DTG-AuxL), which involves a joint generator and a bi-level optimization strategy. Specifically, the joint generator consists of a feature generator and a label generator, which generates the new auxiliary data and new task based on the existing manually collected data and tasks. The data generation process is conducted in the feature space so that it can accommodate various input types, while the label generator architecture is inspired by the model in recommendation which can accommodate both categorical labels for classification and numeric labels for regression. Moreover, we introduce nonlinear interaction terms in the joint generator, making it more expressive to produce beneficial auxiliary data and tasks. To effectively optimize the joint generator and the task learning model, we propose the bi-level optimization strategy with instance regularization. In the lower optimization, the task learning model is optimized by the generated auxiliary data and task. In the upper optimization for the joint generator, we not only utilize the implicit gradient from the primary loss but also the explicit gradient from the proposed instance regularization, to avoid label generation mode collapse. Extensive experiments show that DTG-AuxL outperforms existing methods in various auxiliary learning scenarios, especially when the manually collected auxiliary data and task are unhelpful to the primary task. We summarize our contributions as follows,

- We propose to simultaneously generate auxiliary data and tasks in a joint manner in auxiliary learning for the first time, to the best of our knowledge.
- We propose DTG-AuxL, a joint data-task generation framework applicable in various auxiliary learning scenarios, containing a joint generator and a bi-level optimization strategy.
- Extensive experiments in various auxiliary learning scenarios demonstrate the superiority of our proposed DTG-AuxL framework. We further analyze when and how DTG-AuxL works to bring performance boost.

## 2   Related Work

**Auxiliary Learning** The most widely adopted way in auxiliary learning is to combine the loss of the primary task and the losses of the auxiliary tasks in a linear way [3, 7, 4, 5], where the linear weights are tuned manually or with HPO methods. More recent works [9, 6, 10] propose to automatically assign dynamic weights to the auxiliary losses. [9] propose to assign weights to each loss based on the cosine gradient similarity between the primary loss and each auxiliary loss. Later work [10] aims to make the weighted auxiliary gradient sum closest to the gradient of the primary loss. [13, 1] utilize the bi-level optimization strategy to learn the auxiliary weights, where [1] even propose to combine the losses not only limited to a linear form, but also a nonlinear form. [11, 14] point out that only considering the task-level information is not sufficient, so they jointly consider the weights of different tasks and different data samples within the same task through a joint selector. As aforementioned, these reweighing methods will easily fail to bring improvement when the chosen auxiliary tasks and data are unhelpful. There are also works [2, 1] that generate

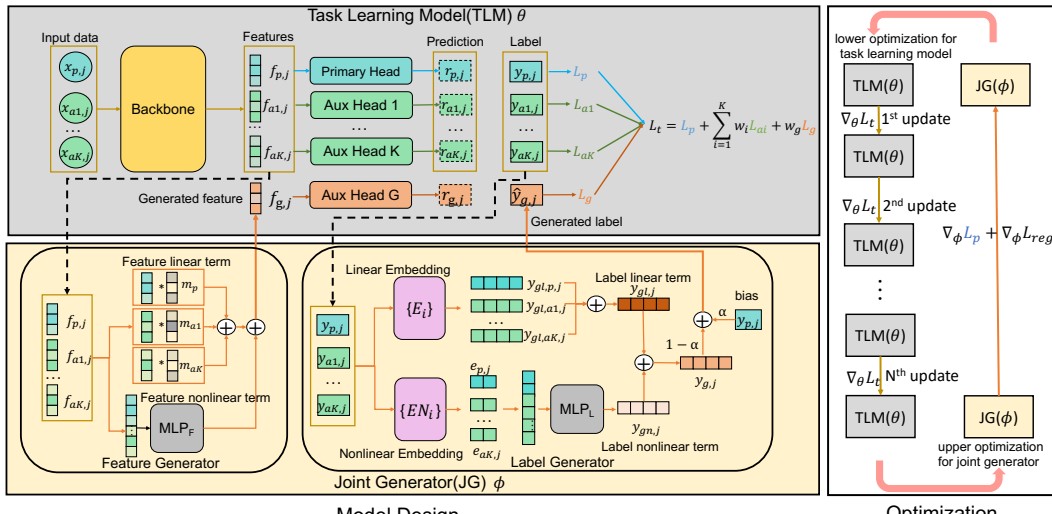

Figure 1: The overall DTG-AuxL framework. In the model design part, the joint generator contains a feature generator and a label generator. The feature generator utilizes the primary and auxiliary features to generate the new auxiliary feature $f_{g,j}$, whose label $\hat{y}_{g,j}$ is generated by the label generator by combining the information of all the auxiliary and primary labels. The optimization part shows that we optimize the task learning model and the joint generator in an alternating bi-level manner.

fine-grained classification auxiliary tasks for the primary classification task. However, they can only be applied to a classification problem. Additionally, they only generate auxiliary tasks without new data, limiting their performance especially when the data of the primary task is inadequate, which is an often encountered scenario in auxiliary learning [9, 1].

**Multi-task learning** Multi-task learning aims to share information among tasks to improve model performance. However, the goal of multi-task learning is to obtain good performance for all the tasks, while auxiliary learning only focuses on the primary task. Multi-task learning methods can be categorized into three parts [15]: multi-task architecture design [16, 17], multi-task optimization [18, 19] and multi-task relationship learning [20], where the multi-task optimization methods involve techniques for optimizing several losses, like loss reweighing [18] and gradient modulation [21] .

## 3 The Proposed Method

The overall DTG-AuxL framework is shown in Figure 1. Next, we give the problem formulation, describe the designs of the joint generator, and present the proposed bi-level optimization strategy.

### 3.1 Preliminaries and Problem Formulation

In auxiliary learning, we have one primary task $T_p$, and totally $K$ auxiliary tasks $\{T_{ai}\}_{i=1}^{K}$. Each of these tasks has its corresponding training dataset, including the primary dataset $D_p = \{(x_{p,j}, y_{p,j})\}_{j=1}^{|D_p|}$, and the dataset for each auxiliary task $D_{ai} = \{(x_{ai,j}, y_{ai,j})\}_{j=1}^{|D_{ai}|}$, where $|\cdot|$ denotes the data sample number of the dataset. If the auxiliary tasks share the same input with the primary task, which is a widely encountered and studied scenario in previous works [11, 1, 9, 10], we have $x_{ai,j} = x_{p,j}$ and $|D_{ai}| = |D_p|$. Besides the datasets, we have a task learning model parameterized by $\theta$ which is used to learn all the tasks together. There is also a validation dataset $D_v$ which is used to evaluate the model performance on the primary task. With these notations, the widely adopted training objective of auxiliary learning is:

$$\mathcal{L}_t(\theta) = \mathcal{L}_p(D_p; \theta) + \sum_{i=1}^{K} w_i \cdot \mathcal{L}_{ai}(D_{ai}; \theta), \qquad (1)$$

where $\mathcal{L}_p$ and $\mathcal{L}_{ai}$ indicate the loss functions of the primary and each auxiliary task. Current reweighing methods focus on how to decide $w_i$ so that the task learning model $\theta$ can achieve the best

performance on $D_v$. However, as aforementioned, the terms $\mathcal{L}_{ai}(D_{ai}, \theta)$ are defined by the manually collected auxiliary data and tasks, with no guarantee to bring benefits for the primary task. Therefore, we propose to simultaneously generate the new beneficial auxiliary data and task in a joint manner, with the following training objective:

$$\mathcal{L}_t(\theta, \phi) = \mathcal{L}_p(D_p; \theta) + \sum_{i=1}^{K} w_i \mathcal{L}_{ai}(D_{ai}; \theta) + w_g \mathcal{L}_g(D_g(\phi); \theta), \tag{2}$$

where the last term is the loss on the generated auxiliary data and task. $D_g(\phi) = \{x_{g,j}, \hat{y}_{g,j}\}_{j=1}^{|D_g|}$ is the generated dataset which contains the new data together with the new label defining the new task, and $\phi$ denotes the parameters of the joint generator that are used to generate $D_g$. Since we only care about the performance of the primary task in auxiliary learning, we keep $\mathcal{L}_g$ the same loss function as $\mathcal{L}_p$, which can be cross entropy or MSE loss, etc. Note that we still keep the original auxiliary loss terms in our training objective, so that it can accommodate the scenario where the existing manually collected auxiliary data and tasks are beneficial. The task weights $w_i$ and $w_g$ are also learnable, which will be optimized together with the generator parameters, and we uniformly denote them all as $\phi$ for convenience. Next, we discuss the details of the generator and how we optimize $\phi$ and $\theta$.

### 3.2 Joint Generator Design

The joint generator involves a feature generator to generate new features and a label generator to generate a new task for the new feature, as shown in the Model Design of Figure 1.

#### 3.2.1 Feature Generator

Since we expect that our data generator can tackle different types of data, a neat and elegant solution is to conduct the generation process in the feature space. In the feature space, different types of input data are transformed to vectors, so we can tackle these vectors in a unified way. Specifically, for the input from different tasks $[x_{p,j}, x_{a1,j}, \cdots, x_{aK,j}]$, the task learning model will first map them to the feature space with an encoder which is generally noted as "backbone", and then use different task-specific heads to tackle each of the tasks. We generate the new features based on the features extracted via the backbone. In another word, the input of the feature generator is a feature list $[f_{p,j}, f_{a1,j}, \cdots, f_{aK,j}]$, where $f_{p,j} = f_{backbone}(x_{p,j}; \theta)$ is a $d$-dimension feature. We next describe how the new features are generated.

**Feature linear term.** A natural way to generate new features is to combine different features with linear masks. We assign an individual feature mask to each of the $K+1$ features, i.e., we have a mask list $[m_p, m_{a1}, \cdots, m_{aK}]$, where each mask is a $d$-dimension vector. Then, we denote the subscript set for the task IDs $\{p, a1, \cdots, aK\}$ as $S$, and the linear combination is conducted as follows:

$$f_{gl,j} = \sum_{i \in S} \hat{m}_i * f_{i,j}, \ \hat{m}_i[k] = \frac{e^{m_i[k]}}{\sum_{j \in S} e^{m_j[k]}}, k = 1 \cdots d, \tag{3}$$

where $gl$ indicates "generated linearly", $*$ means the element-wise multiplication, and $\hat{m}_i$ is the normalized mask. $m_i[k]$ is the $k^{th}$ element of the mask $m_i$, and is normalized by softmax with the elements in the same dimension in other masks. This linear combination with normalized mask works as feature selection from all the input features, and then combines them into a new one.

**Feature nonlinear term.** To make the generated features more expressive, we introduce an MLP (Multi-Layer-Perceptron) to capture the nonlinear feature interaction:

$$f_{gn,j} = MLP_F([f_{p,j}; f_{a1,j}; \cdots; f_{aK,j}]), \tag{4}$$

where $gn$ indicates "generated nonlinearly". All the features are concatenated together and the nonlinear feature interactions are modeled by an MLP, whose output dimension is $d$. Finally, the generated feature is the sum of the linear and nonlinear term, i.e., $f_{g,j} = f_{gl,j} + f_{gn,j}$.

#### 3.2.2 Label Generator

The label generator aims to generate a proper label for the feature produced by the feature generator. We use the labels of all the input data as the input of the label generator to make the generated label expressive enough, i.e., the input of the label generator is a list of labels $[y_{p,j}, y_{a1,j}, \cdots, y_{aK,j}]$.

However, this label list may contain both numeric labels (if the corresponding task is regression) and categorical labels (if the corresponding task is classification). How to utilize different types of label information to generate a new label is the key problem. Inspired by the CTR (click-through rate) model in recommendation [22, 23], which predicts the user's preference towards an item also based on both their numeric and categorical features, we design the label generator with both linear terms and deep nonlinear terms similar to those in the CTR model.

**Label linear term.** The linear term models the direct and independent relationship between each input label and the generated label. We simply keep the dimension of the generated label the same as the primary label $y_{p,j}$ and more adaptive ways to choose the dimension can be an interesting future work. Here, we assume that the dimension of $y_{p,j}$ is $m$, where $m = 1$ if the primary task $T_p$ is regression, or $m$ equals to the number of the total categories if $T_p$ is classification. If the primary task is regression, the generated label is also a scalar value for regression, while if the primary task is classification, the generated label is an $m$-dimension probability distribution vector. Specifically, for each task ID $i \in S$, if $T_i$ is a classification task and it has totally $d_i$ categories, there is an embedding table $E_i$ with dimension $d_i \times m$ for this task. We directly map the label $y_{i,j}$ in task $T_i$ to its $m$-dimension embedding space through the embedding table:

$$y_{gl,i,j} = E_i(y_{i,j}). \tag{5}$$

If $T_i$ is a regression task, we follow [23] to map $y_{i,j}$ to its $m$-dimension embedding. Specifically, we also have an embedding table $E_i$ for this task $T_i$, whose dimension is $d_i \times m$, where $d_i = H$ is a hyper-parameter which is fixed to 10 in our experiments. Then $y_{i,j}$ is mapped into the $m$-dimension embedding space as follows:

$$c_{i,j} = Softmax(Linear(y_{i,j})), c_{i,j} \in R^H, y_{gl,i,j} = \sum_{k=1}^{H} c_{i,j}[k] E_i[k], \tag{6}$$

where the numeric label $y_{i,j}$ is linearly transformed to a $H$-dimension vector $c_{i,j}$, which is used to weigh all the embeddings in the embedding table to obtain the final embedding. Then the final label linear term is obtained by the sum of all the input label embeddings $y_{gl,j} = \sum_{i \in S} y_{gl,i,j}$.

**Label nonlinear term.** In addition to the label linear term capturing how each input label independently influences the final generated label, we also propose a nonlinear term to model the influence of more complex label interactions on the generated label. Specifically, we introduce another group of embedding tables for all the input tasks $\{EN_i\}_{i \in S}$, where the ways to obtain the embeddings of the categorical and numeric labels are the same as eq. (5) and eq. (6), respectively. The dimension of $EN_i$ is $d_i \times m_n$, where $d_i$ equals to the number of total categories if $T_i$ is classification. $d_i$ equals to $H$ (i.e.,10) for regression task, and $m_n$ is a hyper-parameter. With these embedding tables, we can map all the input label list $[y_{p,j}, y_{a1,j}, \cdots, y_{aK,j}]$ to an embedding list $[e_{p,j}, e_{a1,j}, \cdots, e_{aK,j}]$. Then the nonlinear label interactions are captured through an MLP:

$$y_{gn,j} = MLP_L([e_{p,j}; e_{a1,j}; \cdots; e_{aK,j}]). \tag{7}$$

The final generated label for the generated feature $f_{g,j}$ is the sum of the linear and nonlinear terms,

$$y_{g,j} = y_{gl,j} + y_{gn,j}. \tag{8}$$

**Label bias term.** Besides the linear and nonlinear terms, we also introduce a label bias term that guides the generated label to possess similar semantic meaning to the label from the target primary task, $y_{p,j}$. As such, we add $y_{p,j}$ as the label bias term as follows,

$$\hat{y}_{g,j} = \alpha y_{p,j} + (1 - \alpha) * norm(y_{g,j}), \tag{9}$$

where $\alpha \in (0, 1)$ is a learnable parameter initialized as 0.5 and $norm(\cdot)$ is the normalization function. If the primary task is classification, $norm(\cdot)$ will be $softmax(\cdot)$ and $y_{p,j}$ will be converted to the one-hot form. If the primary task is regression with range $(a, b)$, then $norm(\cdot)$ is set to be $(b - a)sigmoid(\cdot) + a$. This bias term makes the generated label lie in the same space as the original label, enabling us to more conveniently explore its semantic meaning, which is also verified to improve model performance in our experiments.

### 3.2.3 Discussion about the Share Input Scenario

The proposed generator can generate the new feature by combining the features from the primary and auxiliary tasks. However, in the most widely studied auxiliary learning scenario [9, 1, 11], all the

tasks share the same input data. We only have the training dataset $\{(x_{p,j}, y_{p,j}, y_{a1,j}, \cdots, y_{aK,j})\}_{j=1}^{|D_p|}$. Therefore, we cannot obtain the auxiliary data $[x_{a1,j}, \cdots, x_{aK,j}]$ from the auxiliary tasks to conduct the feature generation. To tackle this problem, we obtain the new auxiliary data by randomly sampling another data sample in the dataset, which is easy to implement with randomly shuffling the training batch to match another sample for the original sample. Specifically, for a sample $(x_{p,j}, y_{p,j}, y_{a1,j}, \cdots, y_{aK,j})$, we consider another sample $(x_{p,j2}, y_{p,j2}, y_{a1,j2}, \cdots, y_{aK,j2})$ in the dataset to provide new auxiliary data information. The input of the feature generator is $[f_{p,j}, f_{p,j2}]$, which are the features extracted by the backbone with $[x_{p,j}, x_{p,j2}]$ as input. When generating new labels, we need to combine all the existing labels of the two samples, i.e., the input of the label generator is $[y_{p,j}, y_{a1,j}, \cdots, y_{aK,j}, y_{p,j2}, y_{a1,j2}, \cdots, y_{aK,j2}]$, and then the label generation process is the same as before. Note that there are two small details: i) During the label embedding process, $y_{ai,j}$ and $y_{ai,j2}$ share the same embedding table, because they both belong to the same task. ii) For the bias term in eq. (9), we now have two labels from the primary task, $y_{p,j}$ and $y_{p,j2}$, and then eq. (9) can be adjusted to:

$$\hat{y}_{g,j} = \alpha(\beta y_{p,j} + (1-\beta)y_{p,j2}) + (1-\alpha) * norm(y_{g,j}), \tag{10}$$

where $\beta \in (0,1)$ is a learnable parameter of the generator. Till now, we have described the designs of the joint generator, which is applicable in various auxiliary learning scenarios. However, the generator has several parameters to be optimized, like the masks and MLPs. We denote all the learnable parameters in the generator together with the task weights $w_i$ in eq. (2) as $\phi$. Next, we will elaborate how we optimize the task learning model parameters $\theta$ and the generator parameters $\phi$.

### 3.3 Optimization Strategy

**Bi-level optimization.** In our whole framework, the task learning model parameters $\theta$ are expected to minimize the loss of all the selected and generated tasks, while the generator parameters $\phi$ aim to make $\theta$ achieve the best performance on the primary task. These two different goals give rise to the following bi-level optimization problem:

$$\phi^* = \arg\min_{\phi} \mathcal{L}_p(\theta^*(\phi); D_v),$$
$$s.t. \ \theta^*(\phi) = \arg\min_{\theta} \mathcal{L}_t(\theta, \phi), \tag{11}$$

where $\mathcal{L}_t(\theta, \phi)$ is the objective in eq. (2), and $\mathcal{L}_p(\theta^*(\phi); D_v)$ is the primary task loss of the task learning model on the validation dataset. The lower optimization is easy, we can directly obtain the gradient of $\theta$ as $\nabla_{\theta}\mathcal{L}_t(\theta, \phi)$. However, in the upper optimization, $\mathcal{L}_p(\theta^*(\phi); D_v)$ directly relies on $\theta$ instead of $\phi$. Assuming that the Hessian $\nabla_{\theta}^2\mathcal{L}_t(\theta^*(\phi), \phi)$ is positive-definite, we can use the implicit theorem to obtain its implicit gradient $\nabla_{\phi}\mathcal{L}_p(\theta^*(\phi); D_v)$,

$$\nabla_{\phi}\mathcal{L}_p(\theta^*(\phi); D_v) = -\nabla_{\theta}\mathcal{L}_p \cdot (\nabla_{\theta}^2\mathcal{L}_t)^{-1} \cdot \nabla_{\phi}\nabla_{\theta}\mathcal{L}_t|_{(\phi, \theta^*(\phi))}. \tag{12}$$

Detailed derivation can be found in Appendix 1. Since the inverse of the Hessian is often intractable, we follow [24] to use truncated Neumann series to approximate it. $(\nabla_{\theta}^2\mathcal{L}_t)^{-1} \approx \sum_{i=0}^{n}(I - \nabla_{\theta}^2\mathcal{L}_t)^i$, and $n$ is fixed to 3 in all our experiments. Thus, the complete implicit gradient is approximated as:

$$\nabla_{\phi}\mathcal{L}_p(\theta^*(\phi); D_v) \approx -\nabla_{\theta}\mathcal{L}_p \cdot \sum_{i=0}^{n}(I - \nabla_{\theta}^2\mathcal{L}_t)^i \cdot \nabla_{\phi}\nabla_{\theta}\mathcal{L}_t. \tag{13}$$

**Instance regularization.** In the upper optimization, we additionally introduce an instance regularization for the parameterized label $y_{g,j}$ in eq. (8) as follows, to prevent generation mode collapse.

$$\mathcal{L}_{reg}(\phi; D_g) = \begin{cases} \sum_{j=1}^{|D_g|}\sum_{j' \neq j}cos(y_{g,j}, y_{g,j'}) & categorical \\ -\sum_{j=1}^{|D_g|}\bar{y}_{g,j}log(\bar{y}_{g,j}) & numerical \end{cases} \tag{14}$$

where if the generated label is categorical, we expect that the cosine similarity of different generated labels is small. This regularization means we expect that the generated label can keep its instance-level uniqueness, preventing the generated labels of all the generated features from being the same. Similarly, if the generated label is numerical, since it is 1-dimension, we use the entropy regularization to achieve this goal, where $\bar{y}_{g,j} = e^{y_{g,j}} / \sum_{j'=1}^{|D_g|}e^{y_{g,j'}}$. Then in the upper optimization, the gradient of $\phi$ is the sum of the implicit gradient and explicit gradient from the regularization:

$$\nabla_{\phi} = \nabla_{\phi}\mathcal{L}_p(\theta^*(\phi); D_v) + \nabla_{\phi}\mathcal{L}_{reg}(\phi; D_g). \tag{15}$$

Now the gradients of $\theta$ and $\phi$ are obtained, and we follow previous works [1, 11] to alternatingly update $\theta$ and $\phi$. The update loop will continue until convergence, where in each loop we first update $\theta$ for $N$ times, and then update $\phi$ using the upper gradient eq. (15) as shown in Figure 1. $N$ is the interval between two upper updates. We summarize the complete algorithm in Appendix 2, where we do not require an additional validation dataset $D_v$ to calculate the upper objective but reuse the training primary set $D_p$ as done in [13, 14].

## 4 Experiments

### 4.1 Experimental Setup

**Task and Dataset.** We conduct our experiments on two scenarios to validate the generalization ability of our method. One is the most widely studied scenario in previous auxiliary learning works, where the auxiliary tasks share the same input as the primary task (share input). The other one is that the inputs of the primary and auxiliary tasks are different (different input). In the **Share Input** setting, **(i) CUB** [25]: we follow previous works [1, 11] to use the bird visual attribute classification (e.g., whether the bird has white belly) to help the bird species classification primary task, where both the auxiliary and primary labels are categorical. **(ii) CIFAR100** [26]: it is a widely adopted image classification dataset, where there are totally 100 categories. Additionally, each image has a coarse class, e.g., a car belongs to the "vehicles 1" coarse class. We use the coarse classification as the auxiliary task to help the 100-classification primary task. **(iii)** Besides the classification problem, we also focus on the regression problem, where we follow previous works [11, 12] to regard the rating prediction in recommender system as the primary task, and the CTR prediction as the auxiliary task. The primary task is regression and the auxiliary task is binary classification. We choose the widely used Amazon **Toys** and **Movies** [27] datasets, where we use the user ID, item ID and item category as the input data. In the **Different Input** setting, **(i) CIFAR10-100** is a setting where our primary task is the CIFAR10 classification problem, and the auxiliary task is the CIFAR100 classification problem. **(ii) Pet-CUB** is a similar setting where our primary task is fine-grained pet classification on the Pet [28] dataset, while our auxiliary task is the bird species classification in the previously mentioned CUB dataset. More detailed data information is presented in Appendix 4.

**Baselines.** We compare with SOTA auxiliary learning methods, including the reweighing methods and the auxiliary task generation method. Single task learning(STL) is a natural baseline where we only train on the primary task. Equal is a baseline where we assign equal weights 1.0 to all the tasks. Uncert [18] is a dynamic weighting method for multi-task learning based on task uncertainty. GCS [9] and AuxL [1] dynamically reweight the auxiliary losses, and JTDS [11] not only reweighs the tasks but also each data sample within each task. MAXL [1] is a method that automatically generates a fine-grained auxiliary task for the primary classification task. We provide detailed differences between our work and the baselines in Appendix 3.1.

**Implementation details.** In CUB dataset, we respectively adopt ResNet18 [29] and ResNet50 as our backbone. In CIFAR100, we respectively adopt ResNet18 and a 4-layer ConvNet composed of Convolution, Batch Normaliztion and Relu layers as our backbone. In Amazon Toys and Movies, we adopt AutoINT [30] as the backbone. In CIFAR10-100, the backbone is the 4-layer ConvNet and in Pet-CUB the backbone is ResNet18. For the head of each task, we adopt Multi-Layer-Perceptron(MLP) whose layer is searched from $\{1, 2\}$. In the generator, the embedding dimension $m_n$ is searched from $\{32, 64\}$, and the layer number of the MLP is searched from $\{2, 3, 4\}$. More details are presented in Appendix 4.

### 4.2 Experimental Results

#### 4.2.1 Method Performance

**Main results.** Table 1 presents the overall experimental results, showing that our proposed method consistently outperforms existing methods on the diversified auxiliary learning scenarios. During training in the CUB dataset and Pet-CUB dataset, different from [11] that does not use the learning rate scheduler, we found that applying a learning rate scheduler to all the baselines can obtain better or at least on-par performance. Therefore, we report the results with the scheduler. From the results, we have the following observations. **(i)** As expected, our proposed method brings benefits for the primary task when the original manually collected auxiliary data and task are unhelpful. For example,

Table 1: Overall performance. We report the results over 5 random seeds. Note that JTDS cannot handle the different input scenarios and MAXL cannot generate auxiliary tasks for the primary regression task. The metric for classification is accuracy(Acc) and the metric for the rating regression in recommendation is RMSE. Higher accuracy and lower RMSE indicate better results.

| Dataset | CUB | | CIFAR100 | | Toys | Movies | CIFAR10-100 | Pet-CUB |
|---|---|---|---|---|---|---|---|---|
| Metric | Acc(%)↑ | | Acc(%)↑ | | RMSE↓ | RMSE↓ | Acc(%)↑ | Acc(%)↑ |
| Backbone | ResNet18 | ResNet50 | ConvNet | ResNet18 | AutoINT | AutoINT | ConvNet | ResNet18 |
| STL | $76.16_{0.70}$ | $80.46_{0.42}$ | $50.35_{0.50}$ | $55.79_{0.22}$ | $0.9188_{0.0005}$ | $1.0456_{0.0008}$ | $79.35_{0.41}$ | $69.47_{0.16}$ |
| Equal | $74.25_{0.17}$ | $78.28_{0.23}$ | $49.56_{0.31}$ | $56.42_{0.05}$ | $0.9213_{0.0004}$ | $1.0459_{0.0009}$ | $78.99_{0.42}$ | $67.15_{0.41}$ |
| Uncert | $73.94_{0.20}$ | $77.03_{0.24}$ | $48.90_{0.10}$ | $56.95_{0.31}$ | $0.9171_{0.0009}$ | $1.0495_{0.0018}$ | $80.13_{0.17}$ | $63.58_{0.13}$ |
| GCS | $74.11_{0.29}$ | $78.54_{0.54}$ | $49.26_{0.35}$ | $56.57_{0.21}$ | $0.9224_{0.0003}$ | $1.0459_{0.0018}$ | $79.59_{0.52}$ | $67.31_{0.89}$ |
| AuxL | $75.39_{0.59}$ | $78.00_{0.55}$ | $49.39_{0.82}$ | $57.14_{0.25}$ | $0.9186_{0.0005}$ | $1.0483_{0.0019}$ | $79.69_{0.41}$ | $66.17_{0.37}$ |
| JTDS | $76.50_{0.47}$ | $79.34_{0.19}$ | $49.00_{0.33}$ | $57.28_{0.20}$ | $0.9187_{0.0004}$ | $1.0504_{0.0023}$ | - | - |
| MAXL | $75.79_{0.45}$ | $78.48_{0.88}$ | $49.71_{0.37}$ | $56.32_{0.23}$ | - | - | $80.02_{0.51}$ | $68.48_{0.85}$ |
| ours | $\mathbf{77.75_{0.27}}$ | $\mathbf{81.73_{0.20}}$ | $\mathbf{50.94_{0.05}}$ | $\mathbf{57.84_{0.20}}$ | $\mathbf{0.9153_{0.0004}}$ | $\mathbf{1.0426_{0.0009}}$ | $\mathbf{80.64_{0.12}}$ | $\mathbf{70.48_{0.28}}$ |

Table 2: Acc(%) on CUB Fewshot Experiments with ResNet50.

| Method | STL | Equal | Uncert | GCS | AuxL | JTDS | MAXL | ours |
|---|---|---|---|---|---|---|---|---|
| 5 shot | $44.99_{1.15}$ | $45.37_{0.61}$ | $45.63_{0.08}$ | $45.68_{0.67}$ | $45.07_{1.31}$ | $46.13_{0.89}$ | $45.47_{1.06}$ | $\mathbf{52.33_{1.36}}$ |
| 10 shot | $63.25_{0.68}$ | $60.58_{0.81}$ | $59.30_{0.55}$ | $60.81_{0.74}$ | $63.79_{0.11}$ | $63.39_{0.18}$ | $61.70_{0.10}$ | $\mathbf{67.68_{0.33}}$ |

in the CUB dataset with ResNet50 backbone, Movies, and Pet-CUB, the three scenarios, all the methods that utilize auxiliary tasks perform worse than the STL baseline, indicating that the manually chosen auxiliary data and tasks are not beneficial to the primary task. This phenomenon is also observed in previous works [1, 12]. However, our proposed method can utilize the information in the originally harmful auxiliary task to generate new and beneficial auxiliary data and task, so that the model performance can be significantly improved. The ability of our method to convert originally harmful auxiliary labels into beneficial forms is further validated in Appendix 5.2. Although MAXL generates a new fine-grained classification task for the primary task, it brings no improvement in these settings. This is possibly because it requires a hierarchical dataset structure in the primary task. However, in the CUB and Pet-CUB setting, the primary task itself is a fine-grained image classification problem and a more fine-grained meanwhile beneficial task is hard to be automatically generated. Additionally, it does not consider generating new data during generation. **(ii)** When the original auxiliary task is beneficial to the primary task, our proposed method inherits these benefits and can bring further improvement. For example, in CIFAR100 with ResNet18 as backbone and CIFAR10-100 settings, almost all the methods that utilize the auxiliary information outperform the STL baseline, and our proposed method outperforms all the existing auxiliary methods. **(iii)** Whether the auxiliary data and task are beneficial to the primary task depends on the backbone we choose. In the CIFAR100 dataset, when we the use ConvNet backbone, the original auxiliary data and tasks are harmful because all the existing auxiliary methods perform worse than STL, but when we use ResNet18 as the backbone, the auxiliary data and tasks are beneficial.

**Fewshot results.** Since previous works [1, 11] indicate auxiliary learning is more effective when the data of the primary task is inadequate, we conduct experiments on CUB where we only use 5/10 shots of each category for the primary task in Table 2. We can see that when the data of the primary task is 5 shot, all methods that utilize the auxiliary task perform better than STL, while in Table 1, when the primary task has full data, the auxiliary data and task are harmful, indicating whether the auxiliary information is beneficial or not is related to the primary dataset scale. Among all the auxiliary learning methods, our proposed method brings the most significant improvement. Compared to the full dataset setting, the improvement of our method becomes more significant in the fewshot setting. The improvement largely results from that we not only generate new tasks but also new data. The effectiveness of generating new data is further validated in Appendix 5.3.

### 4.2.2 Ablation Studies

To further understand how our proposed method works, we provide the following ablation studies.

Table 3: Generation visualization. The first two rows from Pet-CUB and the last two rows are from CUB. The first and second columns are the images from the primary and auxiliary tasks used to generate new feature and label. The third column is the generated label, where the x-axis represents the category and the y-axis represents the probability of the generated feature belonging to this category. In the fourth column, we visualize the images with maximum probabilities (largest peaks) in the generated label, where besides the image from the peak guided by the bias term, we visualize images from the largest 2 peaks among the small peaks.

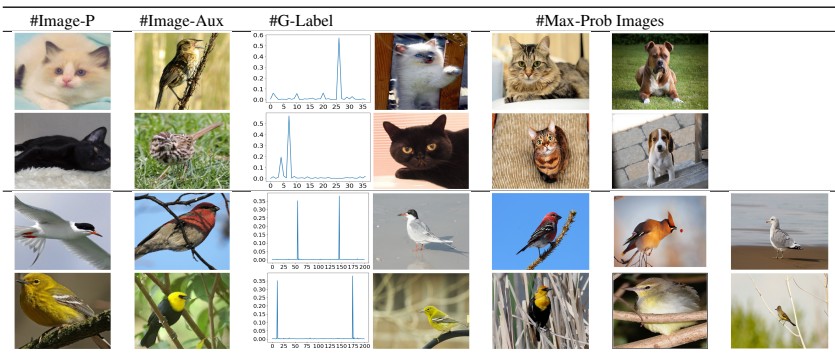

**Generation visualization.** We visualize the generation process in Table 3. The first two rows are from the Pet-CUB dataset in the **Different Input** setting, and the last two rows are from the CUB dataset in the **Share Input** setting. In Pet-CUB, we can see the generated label has one obvious peak from the label bias term, which represents the original category of the primary image. There are also some other small peaks. Specifically, from the first row, we can see that when our feature generator combines a white cat and a yellow-and-black bird, the label generator thinks that the label of the generated feature has the maximum probability of being the original white cat category, the second largest probability of being the yellow-and-black cat category and then a similar color dog. The pattern in the second row is similar to the first row, the texture and color of the bird are combined with the black cat, so that the label generator thinks the generated feature should first belong to the original black cat, then the brown cat with stripes from the bird, and then a similar color dog. In CUB, the generated label has two obvious peaks, which represent the two categories of the primary and auxiliary images. Also, there are some other small peaks, we can see that the images from the small peaks are similar to the primary and auxiliary images, which have reasonable semantic meanings.

**Effectiveness of the framework design.** Our whole framework contains the generator and the optimization strategy. We validate their effectiveness in Table 4. To make the generator expressive, we incorporate nonlinear terms in the generator, and we also incorporate the label bias term in the label generator. To optimize the generator, we propose the bi-level optimization strategy with instance regularization in the upper update. We respectively remove each of these components and observe performance degradation, where w/o F-nonlinear, w/o L-nonlinear, w/o L-bias, w/o bi-level and w/o instance reg respectively represents the variant where we remove the feature nonlinear term, label nonlinear term, label bias term, the overall upper update, and the instance regularization. We find that the non-linear terms indeed help to generate more beneficial data and tasks by making the generator more expressive. Also, the instance regularization is very important in the bi-level optimization. Furthermore, we observe label generation mode collapse without the regularization in Appendix 5.4.

**Comparison with MixUp** During the generation visualization, we find that in the CUB setting, the generation process behaves like MixUp where the inputs are combined in a linear way and the labels are also combined in a linear way. Therefore, we compare our proposed method with different MixUp methods in Table 5. MixUp [31] means we directly utilize the data and labels generated by MixUp as auxiliary tasks. F-MixUp [32] means the linear combination does not happen in the input, but in the manifold feature space. Auto-Mix/Auto-F-Mix means the linear weight of MixUp/F-MixUp is learned automatically with the bi-level optimization, and the task weights are also learnable, i.e, utilizing learnable MixUp/F-MixUp to replace our joint generator. We can see that our proposed method outperforms these MixUp methods. The reasons are quite obvious, as shown in Table 3, our proposed method not only has the two peaks in MixUp, but also other semantically reasonable small peaks. Compared to MixUp, the advantages of our proposed method are three-fold. **(i)** Our method

Table 4: Effectiveness of framework designs. The first three rows explore the effectiveness of the generator components, while the rest show the effectiveness of the training strategies.

| Dataset | CUB | Pet-CUB | Toys |
|---|---|---|---|
| Metric | Acc(%)↑ | Acc(%)↑ | RMSE↓ |
| Backbone | ResNet50 | ResNet18 | AutoINT |
| w/o F-nonlinear | $81.22_{0.27}$ | $67.76_{0.17}$ | $0.9195_{0.0021}$ |
| w/o L-nonlinear | $80.96_{0.37}$ | $66.58_{1.12}$ | $0.9213_{0.0026}$ |
| w/o L-bias | $79.28_{0.34}$ | $69.36_{0.37}$ | $0.9204_{0.0011}$ |
| w/o bi-level | $80.32_{0.42}$ | $68.53_{0.49}$ | $0.9163_{0.0008}$ |
| w/o instance reg | $80.44_{0.34}$ | $67.53_{0.95}$ | $0.9174_{0.0012}$ |
| complete | $\mathbf{81.73}_{0.20}$ | $\mathbf{70.48}_{0.28}$ | $\mathbf{0.9153}_{0.0004}$ |

can not only combine information from the label of the same task but also label from other tasks. **(ii)** Our method can not only model linear relations in generation but also non-linear relations. **(iii)** Our method can handle the Different Input scenario, while MixUp cannot.

To further show the advantage of our proposed generation compared to MixUp, we apply Auto-F-Mix on top of auxiliary learning methods (AuxL, JTDS, MAXL) and the experimental results are presented in Table 6. The results show that both directly combining auxiliary weighting methods(AuxL, JTDS) with Auto-F-Mix and combining the current auxiliary generation method (MAXL) with Auto-F-mix are not as effective as our method, indicating the advantage of our proposed joint generation.

Table 5: Comparison with MixUp Methods. Note that MixUp cannot be applied to the recommendation scenario where the input are categorical features.

| Dataset | CUB | Movies |
|---|---|---|
| Metric | Acc(%)↑ | RMSE↓ |
| Backbone | ResNet18 | AutoINT |
| MixUp | $75.88_{0.50}$ | - |
| Auto-Mix | $76.53_{0.36}$ | - |
| F-MixUp | $75.33_{0.67}$ | $1.0460_{0.0023}$ |
| Auto-F-Mix | $76.88_{0.52}$ | $1.0464_{0.0003}$ |
| ours | $\mathbf{77.75}_{0.27}$ | $\mathbf{1.0426}_{0.0009}$ |

Table 6: Comparison to the combination of Auto-F-Mix and current auxiliary learning methods. We denote Auto-F-Mix as AFM.

| Dataset | CUB | CUB-5shot | Toys |
|---|---|---|---|
| Metric | Acc(%)↑ | Acc(%)↑ | RMSE↓ |
| Backbone | ResNet50 | ResNet50 | AutoINT |
| AFM | $80.30_{0.57}$ | $46.97_{0.95}$ | $0.9187_{0.0003}$ |
| AuxL+AFM | $79.70_{1.07}$ | $47.84_{1.25}$ | $0.9195_{0.0011}$ |
| MAXL+AFM | $80.76_{0.38}$ | $47.96_{0.89}$ | - |
| JTDS+AFM | $80.52_{0.90}$ | $48.14_{0.87}$ | $0.9189_{0.0013}$ |
| ours | $\mathbf{81.73}_{0.20}$ | $\mathbf{52.33}_{1.36}$ | $\mathbf{0.9153}_{0.0004}$ |

## 5 Conclusion

In this paper, we propose to jointly generate beneficial auxiliary data and tasks for auxiliary learning, so that the primary task can still obtain benefits when the manually collected auxiliary data and tasks are unhelpful. We propose the DTG-AuxL framework with a joint generator and a bi-level optimization strategy, which can be applied in various auxiliary learning scenarios. Future works like designing more adaptive generators and more efficient bi-level optimization algorithms can further improve the generation.

## Acknowledgement

This work was supported by the National Key Research and Development Program of China No. 2020AAA0106300, National Natural Science Foundation of China (No. 62222209, 62250008, 62102222), Beijing National Research Center for Information Science and Technology under Grant No. BNR2023RC01003, BNR2023TD03006, and Beijing Key Lab of Networked Multimedia.

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
