# Appendix of Joint Data-Task Generation for Auxiliary Learning

**Hong Chen[1], Xin Wang[1,2]\*, Yuwei Zhou[1], Yijian Qin[1], Chaoyu Guan[1], Wenwu Zhu[1,2]\***
[1]Department of Computer Science and Technology, Tsinghua University
[2]Beijing National Research Center for Information Science and Technology, Tsinghua
{h-chen20,zhou-yw21,qinyj19,guancy19}@mails.tsinghua.edu.cn
{xin_wang,wwzhu}@tsinghua.edu.cn

## 1 Derivation of the Upper Implicit Gradient.

We provide the derivation of the upper implicit gradient in eq. (12). Since we want to obtain $\nabla_\phi \mathcal{L}_p(\theta^*(\phi))$, and $\mathcal{L}_p$ directly relies on $\theta^*$, it is natural to use the chain rule to obtain $\nabla_\phi \mathcal{L}_p(\theta^*(\phi)) = \nabla_\theta \mathcal{L}_p(\theta^*(\phi))\nabla_\phi \theta^*(\phi)$. The key problem is how to obtain $\nabla_\phi \theta^*(\phi)$. Assuming that $\nabla_\theta^2 \mathcal{L}_t(\theta^*(\phi), \phi)$ is positive-definite, we have the following derivations. Since $\theta^*(\phi)$ is a local-minima point of $\mathcal{L}_t(\theta, \phi)$, its partial gradient w.r.t. $\theta$ naturally equals to 0, i.e.,

$$\nabla_\theta \mathcal{L}_t(\theta^*(\phi), \phi) = 0. \tag{A1}$$

We further calculate the gradient of $\phi$ on both sides of eq. (A1), we obtain:

$$\nabla_\phi \nabla_\theta \mathcal{L}_t(\theta^*(\phi), \phi) + \nabla_\theta^2 \mathcal{L}_t(\theta^*(\phi), \phi)\nabla_\phi \theta^*(\phi) = 0. \tag{A2}$$

Since we assume that $\nabla_\theta^2 \mathcal{L}_t(\theta^*(\phi), \phi)$ is positive-definite, it has its inverse matrix. Then we can multiply its inverse matrix $(\nabla_\theta^2 \mathcal{L}_t(\theta^*(\phi), \phi))^{-1}$ on both sides of eq. (A2) and obtain the following results:

$$(\nabla_\theta^2 \mathcal{L}_t(\theta^*(\phi), \phi))^{-1}\nabla_\phi \nabla_\theta \mathcal{L}_t(\theta^*(\phi), \phi) + \nabla_\phi \theta^*(\phi) = 0. \tag{A3}$$

Therefore, we have:

$$\nabla_\phi \theta^*(\phi) = -(\nabla_\theta^2 \mathcal{L}_t(\theta^*(\phi), \phi))^{-1}\nabla_\phi \nabla_\theta \mathcal{L}_t(\theta^*(\phi), \phi), \tag{A4}$$

and the target gradient $\nabla_\phi \mathcal{L}_p(\theta^*(\phi))$ can be obtained using the chain rule:

$$\nabla_\phi \mathcal{L}_p(\theta^*(\phi)) = -\nabla_\theta \mathcal{L}_p(\theta^*(\phi))(\nabla_\theta^2 \mathcal{L}_t(\theta^*(\phi), \phi))^{-1}\nabla_\phi \nabla_\theta \mathcal{L}_t(\theta^*(\phi), \phi), \tag{A5}$$

which is the result in eq. (12).

## 2 DTG-AuxL Algorithm

We summarize the whole DTG-AuxL algorithm in Algorithm 1, where the lower and upper optimization updates are conducted alternately. We use the batch stochastic gradient optimization for both the lower and upper update. After each $N$ steps of lower updates, we conduct one step of upper update. Note that when calculating the upper objective, we use the $D'_p$, which is a dataset by reordering the training primary dataset $D_p$, to replace the validation dataset $D_v$. The reorder operation can make the used primary data batches in the lower update and upper update different, which fits the idea of generalization where we update the model on a batch of data and expect it to perform well on another batch of data. This strategy makes us free of an additional validation dataset $D_v$ during training and it has also been used by [1].

---

\*Corresponding Authors.

37th Conference on Neural Information Processing Systems (NeurIPS 2023).

---

**Algorithm 1** DTG-AuxL Algorithm

---

**Input:** Dataset $\{D_p, D_{a1}, \cdots, D_{aK}\}$, interval between two upper updates $N$;
**Initialize:** task learning model parameters $\theta$, joint generator parameters $\phi$, $\tau = 0$, $D'_p = reorder(D_p)$;
**while** not converged **do**
    // **lower optimization**
    $b_{p,\tau}, \cdots, b_{aK,\tau} = next(\{D_p, \cdots, D_{aK}\})$, // fetch batch
    use $b_{p,\tau}, \cdots, b_{aK,\tau}$ to generate $b_{g,\tau}(\phi)$,
    calculate training loss $\mathcal{L}_t(\theta, \phi)$ on the batches of the used and generated tasks,
    update $\theta$ using gradient $\nabla_\theta \mathcal{L}_t(\theta, \phi)$
    **if** $(\tau + 1)\%N == 0$ **then**
        // **upper optimization**
        $b'_p = next(D'_p)$, calculate the upper objective $\mathcal{L}_p(\theta(\phi); b'_p)$
        //implicit gradient with n-truncated Neumann series.
        $v = p = \nabla_\theta \mathcal{L}_p(\theta(\phi); b'_p)$;
        **for** $i = 1$ **to** $n$ **do**
            $v = v - v \cdot \nabla_\theta^2 \mathcal{L}_t(\theta; \phi)$
            $p = p + v$
        **end for**
        $\nabla_\phi \mathcal{L}_p(\theta(\phi), b'_p) = -p\nabla_\phi \nabla_\theta \mathcal{L}_t(\theta; \phi)$
        calculate explicit regularization gradient $\nabla_\phi \mathcal{L}_{reg}(\phi; b_{g,\tau})$,
        update $\phi$ using $\nabla_\phi \mathcal{L}_p(\theta(\phi), b'_p) + \nabla_\phi \mathcal{L}_{reg}(\phi; b_{g,\tau})$
    **end if**
    $\tau = \tau + 1$;
**end while**
**Return** $\theta, \phi$

---

Table A1: Training time and memory cost of different methods on the CUB dataset with ResNet50 as the backbone.

| Method | STL | Equal | Uncert | GCS | AuxL | JTDS | MAXL | ours |
|---|---|---|---|---|---|---|---|---|
| **Time cost** | 5h33min | 5h35min | 5h32min | 6h55min | 7h17min | 7h5min | 7h40min | 7h35min |
| **Memory cost** | 7059M | 7059M | 7059M | 7213M | 13051M | 13045M | 13999M | 13557M |

# 3 Discussion and Limitation

We first compare our method with the current methods in detail, and then, we discuss the limitations in our work.

## 3.1 Discussion

The compared methods in this work can be summarized as follows:

*Multi-task weighting methods.* **STL**: It is a natural baseline where we only train on the primary task. **Equal**: It is a multi-task learning method, where we assign an equal weight of 1.0 to the loss of each task. **Uncert** [2]: This method utilizes the uncertainty of each task to weigh the loss, where a higher uncertainty results in a lower weight. The multi-task weighting methods are not specifically designed for auxiliary learning, so the performance of the primary task is not guaranteed to be improved.

*Auxiliary weighting methods.* **GCS** [3]: It is an auxiliary gradient reweighing method, which utilizes the gradient similarity between each auxiliary loss and the primary loss to reweigh the auxiliary gradient on the shared parameters. **AuxL** [4]: It combines the auxiliary losses with a nonlinear neural network, and utilizes bi-level optimization to optimize the nonlinear neural network. **JTDS** [5]: It designs a joint scheduler to predict a weight for each data sample within each auxiliary task, and utilizes bi-level optimization to optimize the joint scheduler. The auxiliary weighting methods only reweigh the losses, and cannot generate new data and tasks, which will easily fail to improve the primary task performance when the auxiliary tasks are not properly chosen.

*Auxiliary task generation method.* **MAXL** [4, 6]: It utilizes bi-level optimization to generate a fine-grained classification auxiliary task for the primary classification task. This method is only

constrained to classification label generation and does not generate new data. It does not learn the task weightings, either.

Then we compare our method with current methods in terms of training cost, generality of design, and network inference cost as follows.

- training cost: We report the training time and memory cost of different methods with ResNet50 on the CUB dataset in table A1. Our method has a similar training cost to previous bi-level optimization auxiliary learning methods(AuxL, JTDS, MAXL).

- generality of design: JTDS can only be applied where different tasks have the same input. MAXL can be only applied to the classification problem. Other methods can be applied to general auxiliary learning settings.

- network inference cost: All these methods only change the way of training, and have the same primary task model for inference. Therefore, they have the same inference cost.

## 3.2 Limitation

Although our proposed method can jointly generate auxiliary data and tasks in an effective way, there are still some limitations. Our optimization strategy is based on the gradient-based bi-level optimization method [7]. Although it shows empirical effectiveness in both previous works [4, 5] and our work, it still lacks provable convergence. Naturally, our method has the common limitation of the bi-level optimization method, which has a higher computational cost than the single-level optimization method. Detailed comparison is presented in table A1, where our proposed method has a similar cost with current bi-level auxiliary learning methods like AuxL and JTDS.

## 4 Experimental Details

We conduct experiments on different scenarios and we provide the detailed dataset information and training details.

All the experiments are implemented using PyTorch 1.10.0 and python 3.10.6 and are conducted on a NVIDIA GeForce RTX 3090 GPU with 24GB of memory. In all the experiments, the generator in our proposed method has the embedding dimension $m_n$ randomly searched from {32,64}, and the MLP layer number is searched from {2,3,4}.

### 4.1 CUB Experiment

**Task and dataset.** There are totally 11788 images of 200 species of birds in the CUB dataset. Each image has labels for the attributes of the bird, like "whether the bird has the red wing ". We regard the bird classification as the primary task, and "whether the bird has white belly" as the auxiliary task. The primary task is a 200 classification problem and the auxiliary task is a binary classification. As officially recommended, we use 5994 images for training, 2897 images for validation and the rest 2897 for test. We follow the literature to crop all the images to size 256 [5, 4]. In the training process, the images will be randomly cropped to $224 \times 224$ followed by horizontally flip and Z-score normalization. In the test process, the 256-size images are center-cropped to 224 followed by Z-score normalization.

**Training details.** We adopt the pretrained ResNet18 and ResNet50 [8] and finetune them as previous works [4, 5, 9]. The hyper-parameters are randomly searched for both our method and the baselines. Specifically, the layer number of the MLP task heads is searched from {1, 2}. The batch size is searched from {32, 64}. To train the model, we adopt Adam [10] optimizer with learning rate searched from {1e-3,1e-4,1e-5}, and we totally train 100 epochs. The cosine annealing scheduler is applied for all the methods. In the upper optimization, we use Adam optimizer with learning rate 1e-2. The interval $N$ between two upper updates is searched from {20, 30}. The truncated number of Neumann series is fixed to 3 as previous works [9, 4]. In the few shot setting, we respectively sample 10/5 data samples for each specie of birds from the training dataset, the interval $N$ between two upper updates is searched from {4, 5}.

## 4.2 CIFAR100 Experiment

**Task and dataset.** The CIFAR100 dataset contains 60,000 images and totally 100 categories. Each image also has a coarse category, for example, if an image belongs to the "car" category, it also has a coarse label "vehicles 1". There are totally 20 meta categories. The primary task is a 100 classification problem and the auxiliary task is a 20 classification problem. We use 25,000 samples for training, 25,000 samples for validation and 10,000 samples for test. During the training process, the images are randomly cropped to size 32 with padding 4, and then are normalized with the Z-score. During test process, the images are directly normalized with the Z-score.

**Training details.** We respectively adopt the ResNet18 and ConvNet as the backbone and train them from scratch. The ConvNet contains 4 layers, each layer is composed of Convolution with channel 32, kernel size 3, stride 1, followed by Batch Normalization and ReLU. The hyper-parameters are randomly searched for both our method and the baselines. Specifically, the layer number of the MLP task heads is searched from $\{1, 2\}$. The batch size is searched from $\{256, 512\}$. To train the model, we adopt SGD optimizer with learning rate searched from $\{0.1, 0.01\}$, and we totally train 200 epochs and use multisteplr scheduler with decay factor 0.1 at the 60/120/160 epoch. In the upper optimization, we use Adam optimizer with learning rate 1e-3. The interval $N$ between two upper updates is searched from $\{20, 30\}$. The truncated number of Neumann series is fixed to 3.

## 4.3 Amazon Toys & Movies Experiment

**Task and dataset.** The Amazon Toys and Movies are recommendation dataset from Amazon product, where each user has its rating towards the item they purchased, from 0 to 5. Our primary task is the rating prediction. The auxiliary task is the CTR prediction, where we regard the rating larger than 3 as click as previous works [9, 5]. That is to say, our primary task is a regression problem, and the auxiliary task is a binary classification problem. We split the dataset into training, validation, test chronologically with ratio 0.8/0.1/0.1. In the Toys dataset, we have 134,044 training samples, 16963 validation samples and 16590 test samples. In the Movies dataset, we have 1,357,893 training samples, 170,221 validation samples and 169,419 test samples.

**Training details.** We adopt the AutoINT [11] as the backbone and train it from scratch. The input contains the user ID, item ID and item category. The hyper-parameters are randomly searched for both our method and the baselines. Specifically, the layer number of the MLP task heads is searched from $\{1, 2\}$. The batch size is searched from $\{256, 512\}$. To train the model, we adopt Adam optimizer with learning rate searched from $\{1e-2, 1e-3, 1e-4\}$, and we totally train 20 epochs. In the upper optimization, we use Adam optimizer with learning rate 1e-2. The interval $N$ between two upper updates is searched from $\{30, 50, 100\}$. The truncated number of Neumann series is fixed to 3.

Note that in the previously mentioned scenarios, we focus on the share input setting, where the auxiliary tasks share the same input with the primary task. This scenario is the most widely studied setting in previous works [3–5]. During feature generation, we will randomly shuffle the training batch to obtain an auxiliary sample for the original sample, so that we can conduct the feature generation and label generation as discussed in Sec 3.2.3. The next two scenarios we focus on the different input setting.

## 4.4 CIFAR10-100 Experiment

**Task and dataset.** The dataset for the primary task is CIFAR10, which contains 60,000 images with 10 categories. We split the dataset into 25,000/25,000/10,000 for training/validation/test. The auxiliary dataset is CIFAR100 which is used before. The primary task is a 10 classification problem, and the auxiliary task is a 100 classification problem. During the training process, the images are randomly cropped to size 32 with padding 4, and then are normalized with the Z-score. During test process, the images are directly normalized with the Z-score.

**Training details.** We use the same ConvNet as in the CIFAR100 experiment as the backbone. The hyper-parameters are randomly searched for both our method and the baselines. Specifically, the layer number of the MLP task heads is searched from $\{1, 2\}$. The batch size is searched from $\{256, 512\}$. To train the model, we adopt SGD optimizer with learning rate searched from $\{0.1, 0.01\}$, and we totally train 200 epochs and use multisteplr scheduler with decay factor 0.1 at the 60/120/160 epoch. In the upper optimization, we use SGD optimizer with learning rate 1e-2. The interval $N$

between two upper updates is searched from {20, 30}. The truncated number of Neumann series is fixed to 3.

## 4.5 Pet-CUB Experiment

**Task and dataset.** The dataset for the primary task is Oxford-IIIT Pet,, which contains 7,349 images with 37 categories of cats and dogs. We split the dataset into 3680/1835/1834 for training/validation/test. The auxiliary dataset is CUB which is used before. The primary task is a 37 classification problem, and the auxiliary task is a 200 classification problem. We crop all the images to size 256. In the training process, the images will be randomly cropped to $224 \times 224$ followed by horizontally flip and Z-score normalization. In the test process, the 256-size images are center-cropped to 224 followed by Z-score normalization.

**Training details.** We use the ResNet18 as the backbone and train from scratch. The hyper-parameters are randomly searched for both our method and the baselines. Specifically, the layer number of the MLP task heads is searched from {1, 2}. The batchsize is searched from {32, 64}. To train the model, we adopt Adam optimizer with learning rate searched from {1e-3, 1e-4, 1e-5}, and we totally train 200 epochs. The cosine annealing scheduler is applied for all the methods. In the upper optimization, we use Adam optimizer with learning rate 1e-3. The interval $N$ between two upper updates is searched from {20, 30}. The truncated number of Neumann series is fixed to 3.

# 5 More Ablation Studies

## 5.1 The architecture choice for nonlinear interactions

We try to use MLP and Transformer architecture to capture the nonlinear interaction in the joint generator. In our method, we finally choose the MLP as the architecture because in the CUB dataset we observe MLP achieves higher accuracy than Transformer as the nonlinear interaction architecture as shown in table A2.

Table A2: Comparison of different nonlinear interaction architecture.

| Dataset | CUB | CUB |
|---|---|---|
| Metric | Acc(%)↑ | Acc(%)↑ |
| Backbone | ResNet50 | ResNet18 |
| MLP | $\mathbf{81.73}_{0.20}$ | $\mathbf{77.75}_{0.27}$ |
| Transformer | $80.70_{0.30}$ | $76.73_{0.33}$ |

## 5.2 Convert harmful auxiliary labels into beneficial ones.

When generating the new auxiliary labels, we not only use the label of the primary task as input, but also use the labels of auxiliary tasks. Whether the auxiliary information is efficiently utilized is one key problem in auxiliary learning. We conduct experiments that remove the auxiliary labels from the input of the label generator and the results are given in Table A3. w/o aux means the variant that removes the auxiliary label information. We choose CUB(ResNet50) and Movies to conduct experiments where the original auxiliary tasks are harmful to the primary task. We find that the complete method outperforms the w/o aux variant. This means the generator that has the auxiliary label information as input is better than that without the auxiliary labels. Although the original auxiliary labels are harmful, our method can convert the auxiliary label information into the beneficial form and brings further improvement. That is to say, our method fully utilizes the additional auxiliary information, which is quite promising.

## 5.3 Importance of generating new data.

One advantage of our proposed method is that we can also generate new data. In previous experiments, we have validated the superiority of our method under different scenarios. Here we want to particularly explore the influence of the data generator. We conduct experiments on the previous CUB, CUB-10

Table A3: Utilization of auxiliary labels.

| Dataset | CUB(ResNet50) | Movies |
|---------|---------------|--------|
| Metric | Acc(%)↑ | RMSE↓ |
| STL | $80.46_{0.42}$ | $1.0456_{0.0008}$ |
| w/o aux | $80.84_{0.69}$ | $1.0442_{0.0003}$ |
| complete | $\mathbf{81.73_{0.20}}$ | $\mathbf{1.0426_{0.0009}}$ |

shot and CUB-5 shot settings with ResNet50 as backbone and results are shown in Table A4. w/o data is the variant where we remove the data generator, only using the labels of the primary image to feed the label generator to generate the new label for the primary image. We can see that generating new data is important for improving the performance of the primary task, especially when the data of the primary task are inadequate, which fits one of the motivations of this work.

Table A4: Effectiveness of data generation.

| ResNet50 | CUB | CUB-10shot | CUB-5shot |
|----------|-----|------------|-----------|
| w/o data | $80.72_{0.43}$ | $63.62_{0.42}$ | $47.50_{0.43}$ |
| complete | $\mathbf{81.73_{0.20}}$ | $\mathbf{67.68_{0.33}}$ | $\mathbf{52.33_{1.36}}$ |

## 5.4 Mode Collapse without Instance Regularization.

We visualize the generation process of the variant without instance regularization in CUB and the results are shown in Table A5. We can see that there are always 3 large peaks in the generated label. Besides the original largest peaks provided by the label bias term, there is always a fixed largest peak and the image from the maximum probability category always keeps in the same category "37" whatever the primary and auxiliary images are. We call this phenomenon mode collapse, i.e., the sum of the learnable linear and nonlinear terms in the label generator degenerates to a fixed peak whatever the input is, which is unreasonable and less informative as an auxiliary task. Instead, our proposed the instance regularization, is effective to make the generated label semantically reasonable as shown in Table 3 and improve the primary task performance as shown in Table 4.

## 5.5 The effectiveness of the linear term design

We conduct ablation studies to show the effectiveness of the linear term in the joint generator. We consider two variants of our model, one is that we remove the linear term (which we denote as "w/o linear"), and the other is we replace the linear term with a nonlinear term (which we denote as "two nonlinear"). The results in table A6 show that our original design with the linear term is more effective. Additionally, as we mentioned in the main manuscript, we design the linear term and nonlinear term for feature interaction inspired by the classic recommendation model [12], where the linear term is described to have good "memorization" ability and the nonlinear term has good "generalization" ability, whose effectiveness has been validated in many recommendation scenarios. An interesting and reasonable view for the two parts is to regard them as a form of "mixture of experts", where the linear term and the nonlinear term can be seen as two different experts. The results show that mixture of experts with different abilities (linear "memorization" + nonlinear "generalization") is more effective than mixture of experts with similar abilities (two nonlinear).

Table A5: Generated labels of w/o instance reg in CUB. The columns have the same meaning as Table 3, except for plotting the image from the maximum probability in the fourth column.

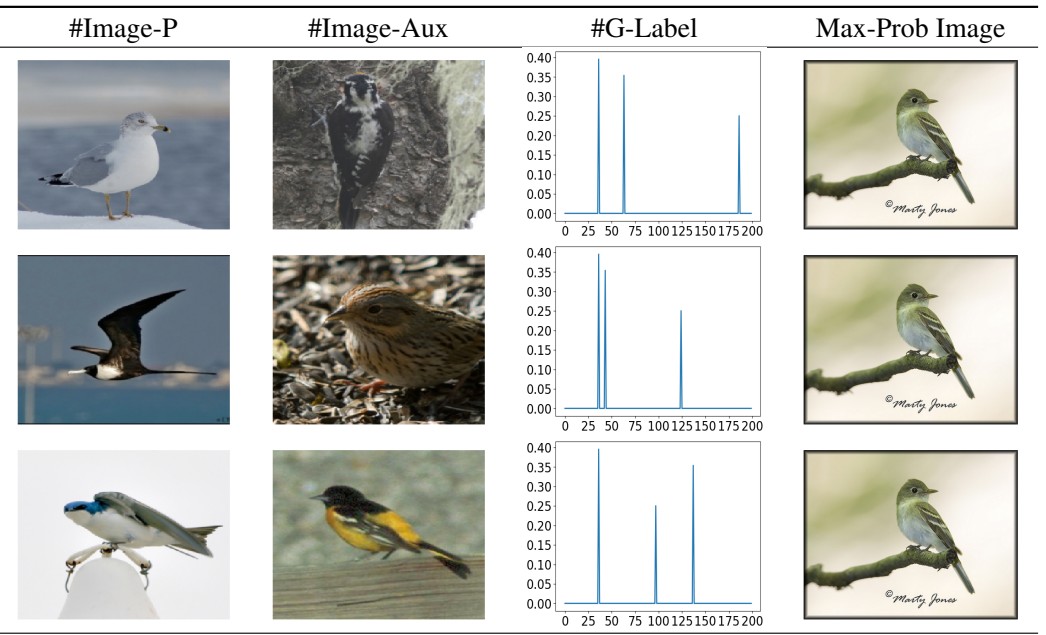

Table A6: The effectiveness of linear term design in joint generator.

| Dataset | CUB | CUB(5 shot) | Toys |
|---|---|---|---|
| Metric | Acc(%)↑ | Acc(%)↑ | RMSE↓ |
| Backbone | ResNet50 | ResNet50 | AutoINT |
| w/o linear | $79.95_{0.24}$ | $49.97_{0.88}$ | $0.9201_{0.0008}$ |
| two nonlinear | $80.44_{0.12}$ | $50.26_{0.76}$ | $0.9182_{0.0011}$ |
| ours | $\mathbf{81.73}_{0.20}$ | $\mathbf{52.33}_{1.36}$ | $\mathbf{0.9153}_{0.0004}$ |

## 5.6 Experiments on segmentation task

We also extend our work to the segmentation task, where we follow [9] to use pretrained EfficientNet as the backbone, and regard segmentation as the main task, normal and depth prediction as the auxiliary tasks. The experiments are conducted on NYUv2 [13] dataset. The results are shown in Table A7. The experiments show that our proposed method can also work well in the segmentation task, demonstrating its generality.

| NYUv2 | STL | MTL | Uncert | GCS | AuxL | MAOAL | ours |
|---|---|---|---|---|---|---|---|
| m-IOU(%) | $32.61_{0.21}$ | $33.15_{0.15}$ | $32.93_{0.32}$ | $32.18_{0.34}$ | $33.06_{0.14}$ | $33.88_{0.23}$ | $\mathbf{34.46}_{0.22}$ |
| pixel-acc(%) | $68.42_{0.62}$ | $68.11_{0.24}$ | $68.74_{0.62}$ | $67.58_{0.80}$ | $68.36_{0.53}$ | $69.17_{0.44}$ | $\mathbf{70.12}_{0.53}$ |

Table A7: Results on segmentation task on NYUv2.

Note that it is not hard to extend our method to NYUv2. First, NYUv2 belongs to the shared input scenarios. Our framework can handle this scenario. Specifically, for each input image, we obtain its feature, and use the input feature with another image's feature to generate a new feature with the feature generator, which is not hard. Additionally, for each image and another image, we have 3 labels for each of them, segmentation map with size (height, width, category_num), depth map with size (height, width, 1) and normal label with size (height, width, 3), when generating new labels, only the final dimension contains label information, therefore, the label generator can generate a (height,

width, category_num)-size new label which is the same size as that of the primary segmentation task. That is to say, the label generator uses different labels in a pixel and uses their linear and nonlinear relations to generate a new label for the same pixel.