# OpenReview forum: "Joint Data-Task Generation for Auxiliary Learning"
_NeurIPS.cc/2023/Conference — NeurIPS 2023 poster_

### Official Review · Reviewer_81dw · 2023-07-05

**Soundness:** 3 good
**Presentation:** 2 fair
**Contribution:** 3 good
**Rating:** 6
**Confidence:** 3

**Summary:**

This paper tackles the problem of auxiliary learning, specifically for cases where the provided auxiliary data actually harms the primary task.
A joint data-task generation framework is proposed, which generates new auxiliary tasks during the optimization process.
The label generator generates data in the feature space and includes both linear and nonlinear interaction terms to produce more expressive data.
Bi-level optimization is used to optimize the entire structure, with instance regularization preventing mode collapse of the generated data.
Experimental results show that the proposed structure can outperform existing methods in a variety of auxiliary learning setups and even prevent performance degradation in certain cases.

**Strengths:**

- The proposed method has strong results in auxiliary learning benchmarks and even prevents performance drop with harmful auxiliary tasks, as shown in the Movies and Pet-CUB experiments. Table 2 also shows its effectiveness in low-data regimes.
- Evaluation and training settings are fully explained in the paper and supplementary material, which helps with understanding and reproducing the results.

**Weaknesses:**

- The idea of bi-level optimization strategy has been proposed and used by prior work [1,2,3]. Thus, this contribution cannot really be claimed by this work. Also, instance regularization is simply an application of the regularization loss introduced in [1].
- Experiments are conducted in limited scale:
  - Most tasks only have a single auxiliary task and the tasks are simple classification or regression problems. Can more auxiliary tasks be used and more complex tasks also be studied as done in previous work, such as on NYUv2 [2]?
  - The extension to dense prediction (e.g., semantic segmentation in NYUv2) seems not so trivial with the proposed framework. It will be beneficial if the authors can demonstrate how this can work with the proposed framework.
- While the improvement even in the presence of harmful auxiliary tasks is a good result, the explanation for why this can be achieved and why previous methods fail is lacking.
  - In section 5.2 of the supplementary material, it seems like even removing auxiliary label information the performance is still superior to STL. Why is this?


[1] Liu, Shikun, Andrew Davison, and Edward Johns. "Self-supervised generalisation with meta auxiliary learning." Advances in Neural Information Processing Systems 32 (2019).
[2] Navon, Aviv, et al. "Auxiliary learning by implicit differentiation." arXiv preprint arXiv:2007.02693 (2020).
[3] Chen, Hong, et al. "Module-Aware Optimization for Auxiliary Learning." Advances in Neural Information Processing Systems 35 (2022): 31827-31840.

**Questions:**

See weaknesses.

Writing:
- The proposed method is akin to MixUp in the feature space, thus I believe the experimental results in section 5.4 in the supplementary material is important and can perhaps be moved to the main paper.
- Experiments:
  - Please add citations for the methods in the experiment tables so it is easier to refer to the corresponding method.
  - Add performance of [3] to Table 1/2.
  - Please fix the formatting of Table 3 and 4, as their current positioning makes this page difficult to read. Table 3 should be a figure and can be column width. The space between Table 4 and the text is so small.

[3] Chen, Hong, et al. "Module-Aware Optimization for Auxiliary Learning." Advances in Neural Information Processing Systems 35 (2022): 31827-31840.

**Limitations:**

The limitations of the work is provided in the supplementary material. I suggest the authors to include at least a short summary in the main paper.

---

> ### Author Rebuttal · Authors · 2023-08-09
>
> Thank the reviewer for the detailed comments and constructive suggestions. We address your concerns point by point as follows.
>
> 1. **About the optimization strategy**: The bi-level optimization has been adopted by previous works[1,2,3], but the instance regularization we adopt is quite different from the regularization in [1] in the following two ways:
>     + [1] calculate the regularization in the class level, which aims to prevent the labels from being a uniform distribution on different classes, but our regularization focuses on the sample level, we want to prevent the labels of all samples from being the same to a fixed point as shown in Table A7 in Appendix 5.5.
>     + The regularization in [1] can be only applied to categorical labels, but our method can be applied to both categorical and numeric labels.
>
> 2. **Extension to NYUv2**: It is not hard to extend our method to NYUv2. First, NYUv2 belongs to the shared input scenarios. Our framework can handle this scenario. Specifically, for each input image, we obtain its feature, and use the input feature with another image's feature to generate a new feature with the feature generator, which is not hard. Additionally, for each image and another image, we have 3 labels for each of them, segmentation map with size (height, width, category_num), depth map with size (height, width, 1) and normal label with size (height, width, 3), when generating new labels, only the final dimension contains label information, therefore, the label generator can generate a (height, width, category_num)-size new label which is the same size as that of the primary segmentation task. That is to say, the label generator uses different labels in a pixel and uses their linear and nonlinear relations to generate a new label for the same pixel. With this principle, we follow [3] to use the EfficientNet as the backbone and the results are shown in the following table,
>
>     |  NYUv2   | SLL  | Equal | HPO-tune | Uncert | GCS| AuxL  |MAOAL | ours|
>     |  ----  | ----  | ----|----  | ----  | ----|----  | ----  | ----|
>     | m-IOU(%) $\uparrow$  |$32.61_{0.21}$  | $33.15_{0.15}$| $33.09_{0.25}$| $32.93_{0.32}$| $32.18_{0.34}$| $33.06_{0.14}$| $33.88_{0.23}$|  $\textbf{34.46}_{0.22}$|
>     | pixel-acc(%) $\uparrow$ | $68.42_{0.62}$ |$68.11_{0.24}$|$68.52_{0.52}$| $68.74_{0.62}$| $ 67.58_{0.80}$|$68.36_{0.53}$| $69.17_{0.44}$| $\textbf{70.12}_{0.53}$|
>
>     The results show that our proposed method outperforms all the baselines, and our method can be applied to this scenario.
>
> 3. **Why our method achieves improvement**: Compared to the existing methods, our advantage lies in the information brought by the generated new data and task. Specifically, current weighting methods limit their performance because the existing auxiliary data and tasks are unhelpful. Current auxiliary generation methods have limited performance because they ignore the data generation and limit their label to fine-grained labels, which may be not proper for all the scenarios.
>     + In Appendix 5.2, we remove the auxiliary labels from the input of the label generator, which means we only use the primary labels to generate the new label (Note that in Figure 1 in the main manuscript, both existing the primary labels and auxiliary labels will be used as the input of the label generator.). The performance of w/o aux is better than STL means that only using the primary labels to generate new auxiliary label, our method can still bring performance improvement. This phenomenon also means our method can be used to generate auxiliary data and tasks even when there are only primary labels but no auxiliary labels.
>
> 4. **Writing**:
>     + We are glad to include section 5.4 in our final version.
>     + We will add citations to the table.
>     + Part of the performance of [3] is listed in the following table and we will include the results on other settings and datasets in our final version.
>
>     | ResNet50    |MAOAL[3]  | ours |
>     |  ----  | ----  | ----|
>     | CUB  |$79.68_{0.45}$  | $81.73_{0.20}$|
>     | CUB-5shot | $45.53_{0.67}$ |$52.33_{1.36}$|
>     + We will make the tables well formatted in our final version.
>     + We will include a summary of the limitations in the main paper in our final version.
>
> Thanks for your suggestions. We believe the experiments in the rebuttal will further strengthen our work and the writing suggestions will improve the readability. We are glad to answer further questions if necessary.
>
>
> [1] Liu, Shikun, Andrew Davison, and Edward Johns. "Self-supervised generalisation with meta auxiliary learning." Advances in Neural Information Processing Systems 32 (2019).
>
> [2] Navon, Aviv, et al. "Auxiliary learning by implicit differentiation." arXiv preprint arXiv:2007.02693 (2020).
>
> [3] Chen, Hong, et al. "Module-Aware Optimization for Auxiliary Learning." Advances in Neural Information Processing Systems 35 (2022): 31827-31840.

---

> > ### Comment · Reviewer_81dw · 2023-08-11
> > **Follow-up by Reviewer**
> >
> > I thank the authors for providing detailed responses to my concerns, and I have read through all the responses to the other reviews as well. The responses adequately address my concerns. Considering all the reviews, I will increase my score from borderline accept to weak accept.

---

> > > ### Author Response · Authors · 2023-08-14
> > > **Thanks for the follow-up**
> > >
> > > We thank the reviewer for the detailed check and response to our rebuttal content, and we believe this fruitful rebuttal further improves our paper.

---

### Official Review · Reviewer_Gctb · 2023-07-06

**Soundness:** 2 fair
**Presentation:** 3 good
**Contribution:** 3 good
**Rating:** 5
**Confidence:** 2

**Summary:**

This paper focuses on Auxiliary Learning, a technique aiming to improve the model generalization ability on the primary task with the help of related auxiliary tasks, and proposes a novel joint data-task generation framework for auxiliary learning (DTG-AuxL). DTG-AuxL is designed to mitigate certain issues present in existing methods. Specifically, existing methods necessitate the presence of beneficial information in the auxiliary data and tasks; otherwise, the useless auxiliary information will do harm to the primary task when the involved auxiliary data and tasks are improperly collected. DTG-AuxL features a joint generator and a bi-level optimization strategy and can simultaneously generate auxiliary data and tasks in a joint manner. Extensive experiments show that DTG-AuxL outperforms existing methods in various auxiliary learning scenarios.

**Strengths:**

This paper proposes DTG-AuxL, a joint data-task generation framework applicable in various auxiliary learning scenarios. DTG-AuxL contains a joint generator and a bi-level optimization strategy and can simultaneously generate auxiliary data and tasks in a joint manner, which is the first attempt to do so.

DTG-AuxL can adapt to different kinds of input data, such as tabular data and visual images. Because the data generation process is conducted in the feature space and the label generator can accommodate both categorical labels for classification and numeric labels for regression. Consequently, DTG-AuxL has wide applicability.

How to effectively optimize the parameters in the generation framework is a challenging problem to guarantee the generated data and task are beneficial to the primary task. This paper proposes a bi-level optimization strategy as well as instance regularization to tackle this challenge.

Extensive experiments on two scenarios validate the generalization ability of DTG-AuxL. The first scenario is that the auxiliary tasks share the same input as the primary task, and the second scenario is that the inputs of the primary and auxiliary tasks are different. Experimental results show that DTG-AuxL outperforms existing methods on diversified scenarios, including few-shot learning scenarios. Specifically, DTG-AuxL is able to convert originally harmful auxiliary labels into beneficial forms.


**Weaknesses:**

Table 3 and Table 4 is not appropriate.

This paper conducts experiments on the following scenarios:

1. Share Input: visual inputs & classification task (primary) + visual inputs & classification task (auxiliary)
2. Share Input: tabular inputs & regression task (primary) + tabular inputs & classification task (auxiliary)
3. Different Input: visual inputs & classification task (primary) + visual inputs & classification task (auxiliary)

But in order to verify the broad applicability of DTG-AuxL in the paper, more different types of experiments need to be done, such as Different Input: tabular inputs & classification task (primary) + tabular inputs & classification task (auxiliary).

Finally, I am a little confused, does the so-called task generation refer to label generation? Why can't label be regarded as part of data? Shouldn't task generation literally include loss functions or other ingredients?


**Questions:**

See my questions in Weaknesses.

**Limitations:**

No comments.

---

> ### Author Rebuttal · Authors · 2023-08-09
>
> Thank the reviewer for the detailed comments and constructive suggestions. We address your concerns point by point as follows.
>
> 1.  **About Table 3 and Table 4**: We will make them better formatted in our final version. Thanks for your suggestion.
>
> 2. **More experiments**: We follow the reviewer's suggestions to conduct experiments on the setting Different Input: Tabular inputs, primary classification + auxiliary classification.
>
>     Specifically, we consider the cross-domain CTR prediction in recommendation, i.e., using the user behaviors in the Movies domain to help the target Toys domain CTR prediction, and evaluate the results with AUC metric(higher indicates better). The results are shown as follows,
>
>     |   Method  | STL  | Equal | Uncert | GCS | AuxL | JTDS | MAXL| ours |
>     |  ----  | ----  | ----|----  | ----|----| ----| ----| ----|
>     |  AUC $\uparrow$ | $0.6508_{0.0053}$| $0.6486_{0.0071}$| $0.6685_{0.0050}$ | $0.6661_{0.6664}$| $0.6513_{0.0058}$|__|$0.6628_{0.0049}$|$\textbf{0.6787}_{0.0068}$|
>
>    Additionally, we follow [1] to conduct experiments on the NYUv2 dataset to use the depth prediction and normal prediction to help the image segmentation, which is share input, image data, primary classification, and auxiliary regression, and the results are shown as follows (Note that JTDS and MAXL cannot be applied to this scenario),
>
>     |  NYUv2   | SLL  | Equal | HPO-tune | Uncert | GCS| AuxL  |MAOAL | ours|
>     |  ----  | ----  | ----|----  | ----  | ----|----  | ----  | ----|
>     | m-IOU(%) $\uparrow$  |$32.61_{0.21}$  | $33.15_{0.15}$| $33.09_{0.25}$| $32.93_{0.32}$| $32.18_{0.34}$| $33.06_{0.14}$| $33.88_{0.23}$|  $\textbf{34.46}_{0.22}$|
>     | pixel-acc(%) $\uparrow$ | $68.42_{0.62}$ |$68.11_{0.24}$|$68.52_{0.52}$| $68.74_{0.62}$| $ 67.58_{0.80}$|$68.36_{0.53}$| $69.17_{0.44}$| $\textbf{70.12}_{0.53}$|
>
>     The results show that our proposed method outperforms all the baselines, which further shows the effectiveness of our method. We believe these results can better support our paper. Thanks for your suggestions.
>
> 3. **About task generation:** Yes, we follow previous works [2,3] to call the label generation as task generation, and they generate a new label for the input data and state that they generate an auxiliary task for the input data. As a matter of fact, in many shared-input multi-task learning scenarios, and we have several labels for the input (e.g., in NYUv2, one input image, and the segmentation map, the normal vector and the depth map as the labels). Predicting each of the labels defines a task (e.g., semantic segmentation, depth prediction and normal prediction), and predicting all of them together is called multi-task learning. Considering these literature, we call generating a new label as task generation.
>
> [1] Chen, Hong, et al. "Module-Aware Optimization for Auxiliary Learning." Advances in Neural Information Processing Systems 35 (2022): 31827-31840.
>
> [2] Liu, Shikun, Andrew Davison, and Edward Johns. "Self-supervised generalisation with meta auxiliary learning." Advances in Neural Information Processing Systems 32 (2019).
>
> [3] Navon, Aviv, et al. "Auxiliary Learning by Implicit Differentiation." International Conference on Learning Representations. 2020.

---

### Official Review · Reviewer_iWiK · 2023-07-07

**Soundness:** 3 good
**Presentation:** 3 good
**Contribution:** 3 good
**Rating:** 6
**Confidence:** 4

**Summary:**

This paper studies the problem of learning with auxiliary tasks. Motivated by the observation that in most scenario the auxiliary data is hard to collect, the authors propose to train a model that can automatically generate auxiliary data (and labels) for any task at hand. Specifically, the authors propose a data generator network that generates new data based on the current data, and the network is learned by a bi-level optimization where the final objective is to maximize the performance on the validation set of the primary task. The authors show the advantage of the proposed method on several datasets and settings.

**Strengths:**

- The motivation is clear and sensitive and the proposed method is sound.

- The idea of auxiliary data generation might not be completely new but the authors propose a unified way that can generate data for different tasks and different input/output format, which is a good contribution to me.

- The paper is overall well written and easy to follow.

**Weaknesses:**

- To tackle with tasks with different input format, the authors propose to generate the features instead of the raw data. This may be a problem because in this way only the task-specific heads are trained on the generated data, but the backbone is still only learned on the primary task (as well as the already existed auxiliary tasks). This is suboptimal and might limit the impact of the generated data. Maybe the authors could comment on this?

- In the feature linear interpolation, are the masks (L143) input-independent?

- The instance regularization loss for numerical labels (Equation 14), if the generated labels have one label that is extremely large, and other labels that are the same and extremely small, then the $\bar{y}_{g,j}$ will be 1 for one label and 0 for others, which will minimize the regularization loss. This seems not encouraging the diversity of the labels.

- One question I have is about the visualization of the generated data (Table 3). I wonder where do the Max-Prob Images come from? Are they the images that have the maximal probability (predicted by the trained model) on the label? Have the authors tried to visualize the images in the training set which have the feature that is closest to the generated feature, i.e., using the generated feature to do a knn search in the training set?

- Have the authors tried the setting where there is no ready-to-use auxiliary tasks and only the primary task is available?


===========

Post rebuttal: The rebuttal partially address my concerns and I will keep my score of weak accept.

**Questions:**

See Weaknesses.

---

> ### Author Rebuttal · Authors · 2023-08-09
>
> Thank the reviewer for the detailed comments and constructive suggestions. We address your concerns point by point as follows.
> 1. **Reply to Weakness 1**:
>
>     **The impact of generated data and task on the backbone**: The generated data not only influences the task-specific head but also the backbone. Note that in Figure 1, the loss of the generated auxiliary task relies on the generated feature and label, and the generated feature is obtained from the existing features through the feature generator, and the existing features are extracted using the backbone. Therefore, the loss of the generated auxiliary task will also have its gradient on the backbone, and improves the performance of the backbone on the primary task. Thanks for your careful reading and thinking, and we will add one paragraph to explain it in our final version.
>
> 2. **Reply to Weakness 2**:
>
>     **Whether the mask(L143) is input-independent**: Yes, in our implementation, we use a global mask for all the data and find it works well for various scenarios, but it will be an interesting problem whether designing some sample-specific masks can further improve the performance in our future work.
>
> 3. **Reply to Weakness 3**:
>
>     **Instance regularization for numerical labels**: If only using the regularization, the generated label will be not diverse as the reviewer pointed out. However, there is also the implicit gradient to make the generated label helpful to the primary task. Both the implicit gradient and the explicit gradient working together can cause a beneficial joint generation as shown in Table 4, without using the instance regularization will do harm to the recommendation performance. Additionally, in some other particular scenarios, users may add a hyperparameter to control the impact of the regularization.
>
> 4. **Reply to Weakness 4**:
>
>     **About the visualization**: We find the Max-Prob images with the following process: we first find the largest peak of the generated label, e.g., 7 in the 2nd row in Table 3, and then we will see in the training set what images have label 7, and then put the image of category 7 in Table 3. Then, for the second largest peak which is label 4, we also find its corresponding image in the training set with category 4.
>
>     The generation visualization aims to find whether the generated task (label) has some semantic meanings, and the answer is yes as explained in L330-356. Considering that the generated task and the primary task do not share the same task head (indicating that their classification boundaries for the features are different, and the features of the auxiliary task and the primary task are not in the same space), visualizing the images searched by features may not be that meaningful.
>
> 5. **Reply to Weakness 5**:
>
>     **Whether we try there are no ready-to-use auxiliary tasks**: In Appendix 5.2, we try to remove the auxiliary labels, and for reading convenience, we report the results in the following table (best performance is bolded and the second is underlined). We find that:
>     + Our method without auxiliary labels, ours (w/o aux), outperforms STL and all the baselines, indicating our method can be applied when there are no auxiliary labels.
>     + Our complete method with the auxiliary labels, further improves the performance by converting the information in the auxiliary labels into a beneficial form to the primary task.
>
>     |                      | STL           |    Equal      | Uncert        | GCS           | AuxL          | JTDS          | MAXL       | ours (w/o aux) | ours          |
>     |----------------------|---------------|---------------|---------------|---------------|---------------|---------------|------------|----------------|---------------|
>     | CUB(ResNet50)/Acc(%) $\uparrow$ | $80.46_{0.42}$    | $78.28_{0.23}$    | $77.03_{0.24}$    | $78.54_{0.54}$    | $78.00_{0.55}$    | $79.34_{0.19}$    | $78.48_{0.88}$ | $\underline{80.84}_{0.69}$     | $\textbf{81.73}_{0.20}$    |
>     | Movies/RMSE    $\downarrow$      | $1.0456_{0.0008}$ | $1.0459_{0.0009}$ | $1.0495_{0.0018}$ | $1.0459_{0.0018}$ | $1.0483_{0.0019}$ | $1.0504_{0.0023}$ | -          | $\underline{1.0442}_{0.0003}$  | $\textbf{1.0426}_{0.0009}$ |
>     We are glad to answer further questions if necessary.

---

### Official Review · Reviewer_HDHq · 2023-07-07

**Soundness:** 4 excellent
**Presentation:** 4 excellent
**Contribution:** 3 good
**Rating:** 7
**Confidence:** 5

**Summary:**

Motivated by the manually collected auxiliary data and tasks in auxiliary learning may have negative impacts on the primary task, this paper proposes an auxiliary learning framework that jointly generates new auxiliary data and task. To generate new data for various auxiliary scenarios, the authors design an adaptive feature generator and a label generator, which considers both linear and nonlinear relations. Extensive experiments show that the proposed method can bring benefits to the primary task, even when the manually collected auxiliary data and task are potentially harmful.

**Strengths:**


1.	The paper is well-motivated and tackles an important problem in auxiliary learning. To collect the beneficial auxiliary data and task really needs human trials and domain knowledge.
2.	The paper is well organized with clear discussion about related works, reasonable method formulation, and experiments to support the claims.
3.	The experiments are conducted across different domains and datasets to show the effectiveness and generality of the proposed method.
4.	Some ablations bring interesting insights.

**Weaknesses:**


1.	Since this work focuses on using generation to tackle the auxiliary learning problem, why not use some pretrained generative models like Stable Diffusion? The Stable Diffusion can also use conditional generation which gives similar data as provided by Table 3, and the paper can be strengthen with some discussion about this.
2.	The most advanced models are pretrained, whether the proposed method can be applied to pretrained models is not clear.


**Questions:**

See above.

**Limitations:**

I don't see any concerns regarding the societal impact in this work.

---

> ### Author Rebuttal · Authors · 2023-08-09
>
> Thank you for your careful comments and constructive suggestions. We address your concerns point by point as follows.
>
> 1. **Reply to Weakness1**:
>
>     **Discussion about generation with stable diffusion**: Stable Diffusion is a pretrained text-to-image generation model, and it relies on large datasets to train. It can be used to generate image data and with the text as the labels. Our method is superior to Stable Diffusion in auxiliary learning in the following two ways:
>     + Stable diffusion can be only used to generate auxiliary data for image classification task, but our proposed method can generate auxiliary data and tasks for different types of data, such as image, and tabular data.
>
>     + Even in the image generation scenario, the generated data by Stable diffusion is not necessarily helpful to the primary task, but the bi-level optimization in our work guarantees that the generated data and task are beneficial to the primary task.
>
>     Therefore, our proposed method is quite meaningful in auxiliary learning despite the existence of Stable Diffusion. We will include the discussion in our final version. Thanks for your suggestions.
>
> 2. **Reply to Weakness2**:
>
>     **Whether the proposed method works when the model is pretrained**: In our CUB and CUB-fewshot experiments, the used backbones are pretrained ResNet18 and ResNet50, and we finetune them on CUB dataset, and this detailed experimental setting is given in Appendix 4.1. Additionally, we follow the setting of [1] to use the pretrained EfficientNet to finetune on the NYUv2 dataset to help the semantic segmentation primary task. The results are shown in the following table, which show that our proposed method can work well when conduct auxiliary finetuning. Therefore, our method can work well when applied to pretrained models.
>     |  NYUv2   | SLL  | Equal | HPO-tune | Uncert | GCS| AuxL  |MAOAL | ours|
>     |  ----  | ----  | ----|----  | ----  | ----|----  | ----  | ----|
>     | m-IOU(%) $\uparrow$  |$32.61_{0.21}$  | $33.15_{0.15}$| $33.09_{0.25}$| $32.93_{0.32}$| $32.18_{0.34}$| $33.06_{0.14}$| $33.88_{0.23}$|  $\textbf{34.46}_{0.22}$|
>     | pixel-acc(%) $\uparrow$ | $68.42_{0.62}$ |$68.11_{0.24}$|$68.52_{0.52}$| $68.74_{0.62}$| $ 67.58_{0.80}$|$68.36_{0.53}$| $69.17_{0.44}$| $\textbf{70.12}_{0.53}$|
>
> We are glad to answer further questions if necessary.
>
> [1] Chen, Hong, et al. "Module-Aware Optimization for Auxiliary Learning." Advances in Neural Information Processing Systems 35 (2022): 31827-31840.

---

> ### Comment · Reviewer_HDHq · 2023-08-17
> **Response to the rebuttal**
>
> Thanks to the efforts made by the authors, all my concerns have been addressed. Therefore, I confirm my tendency to accept this work.

---

> > ### Author Response · Authors · 2023-08-17
> >
> > Thank you to the reviewers for the suggestions and response to our work and rebuttal content. We believe with the rebuttal content, our paper will be made more clear.

---

### Official Review · Reviewer_SrwV · 2023-07-07

**Soundness:** 3 good
**Presentation:** 4 excellent
**Contribution:** 3 good
**Rating:** 7
**Confidence:** 5

**Summary:**

This paper proposes an auxiliary learning framework that jointly generates new auxiliary data and new labels, to tackle the negative transfer problem in auxiliary learning. The framework is designed to be generic to various auxiliary scenarios, both multiple input and single input, regression and classification. The bi-level optimization strategy is designed to optimize the task learning model and the joint generation model. Experiments on several benchmarks show that the proposed method outperforms existing methods.

**Strengths:**

1.	The paper is very clear and well-written.
2.	The idea to generate new data and task to avoid the potential negative transfer in manually designed auxiliary tasks is novel and reasonable.
3.	The relations and differences compared to previous works are clearly discussed.
4.	The proposed method is generic and makes sense.
5.	Experiments are well conducted, and the effectiveness of each component is well validated. The visualizations are reasonable and interesting.


**Weaknesses:**

1.	In the single input scenario, the proposed method behaves like MixUp, and I think the comparison with MixUp is important and should be put to the main manuscript instead of the appendix, although I understand the page is limited.
2.	Considering that current large pretrained models usually have a lot of parameters, if we want to finetune these models, as the limitation part discussed, we will need more computational cost. Whether this method can serve the current pretrain-tuning paradigm will largely influences its significance.


**Questions:**

Please refer to the weakness part. Although I think this paper is overall of good quality, I want to know how it can adapt to the current pretrained models.

---

> ### Author Rebuttal · Authors · 2023-08-09
>
> Thank the reviewer for the constructive comments and suggestions. We address your concerns point by point as follows
>
> 1. **Rely to Weakness 1**:
>     **Comparison with MixUp**:
>     We agree with the reviewer that comparison with MixUp is important in the shared-input scenario, so in Appendix 5.4, we conduct extensive experiments and list the advantages of our method as the reviewer pointed out. We are glad to include it into the main manuscript in our final version.
>
> 2. **Reply to Weakness 2**:
>
>     **Applying our method to large pretrained models**: Applying our method to directly finetune all the parameters in large pretrained models will indeed result in high computational cost, which is the limitation of bi-level optimization. However, luckily, there are many parameter-efficient finetuning methods, like LoRA[1], Adapter[2], or even we can choose partial parameters to finetune while fixing most of the pretrained parameters, then the computational cost will be largely reduced, and our method can be easily applied to large pretrained models. It is a very interesting future work and we will try to extend this work to large pretrained models. Thanks for your suggestions.
>
> We are glad to answer further questions if necessary.
>
> [1] Houlsby, Neil, et al. "Parameter-efficient transfer learning for NLP." International Conference on Machine Learning. PMLR, 2019.
>
> [2] Hu, Edward J., et al. "Lora: Low-rank adaptation of large language models." arXiv preprint arXiv:2106.09685 (2021).

---

### Decision · Program_Chairs · 2023-09-21

**Decision:**

Accept (poster)

**Comment:**

This paper studies the problem of learning with auxiliary tasks, specifically where the provided auxiliary data harms the primary task. Based on the observation that auxiliary data is hard to collect in most cases, a joint data-task generation framework (DTG-AuxL) is proposed and the authors design a feature generator and a label generator for various auxiliary scenarios. Extensive experiments successfully validate the effectiveness of the proposed method and some ablations bring interesting insights. Overall, all reviewers agreed with the contribution of this paper and the significance to the community can be further improved if DTG-AuxL is still effective on large pre-trained models.